# Scoreformer: A Surrogate Model For Large-Scale Prediction of Docking Scores

**Álvaro Ciudad** [1]   **Adrián Morales-Pastor** [1]   **Laura Malo** [2 3]   **Isaac Filella-Mercè** [3]   **Victor Guallar** [3 4]   **Alexis Molina** [1 3]

## Abstract

In this study, we present ScoreFormer, a novel graph transformer model designed to accurately predict molecular docking scores, thereby optimizing high-throughput virtual screening (HTVS) in drug discovery. The architecture integrates Principal Neighborhood Aggregation (PNA) and Learnable Random Walk Positional Encodings (LRWPE), enhancing the model's ability to understand complex molecular structures and their relationship with their respective docking scores. This approach significantly surpasses traditional HTVS methods and recent Graph Neural Network (GNN) models in both recovery and efficiency due to a wider coverage of the chemical space and enhanced performance. Our results demonstrate that ScoreFormer achieves competitive performance in docking score prediction and offers a substantial 1.65-fold reduction in inference time compared to existing models. We evaluated ScoreFormer across multiple datasets under various conditions, confirming its robustness and reliability in identifying potential drug candidates rapidly.

## 1. Introduction

Virtual screening (VS) is a key computational technique in drug discovery, utilized for evaluating the potential binding affinity of numerous molecules with target proteins. The main aim of VS is to identify molecules with potential interaction capabilities, thereby streamlining the drug discovery process and reducing the need for extensive, costly exper-

imental assays (Lavecchia & Di Giovanni, 2013). HTVS represents an advanced and more efficient form of VS. Characterized by its ability to rapidly assess vast molecular libraries, HTVS significantly contributes to identifying potential biological interactions.

The evolution of HTVS has been significantly influenced by the advancement of combinatorial chemistry, enabling the generation of extensive and diverse compound libraries. Previously, drug discovery relied on smaller and less diverse libraries, limiting the scope of potential candidates. The integration of combinatorial chemistry with HTVS allows for the utilization of larger compound libraries, alongside advanced computational evaluation techniques. This integration has accelerated the process of identifying suitable drug candidates by enabling rapid and precise assessment of ligand affinity across these vast libraries. Such efficiency is crucial in modern drug discovery as it can considerably shorten the hit-finding phase and enhance the overall hit rate. As a result, there's an increasing need for methodologies that can swiftly and reliably navigate this expanded chemical space (Sadybekov & Katritch, 2023).

In this context, we present ScoreFormer, a graph transformer model developed to accurately predict molecular docking scores, thereby optimizing the efficiency of HTVS. ScoreFormer is designed to interpret molecular structures and their relationship with docking scores. Our rigorous evaluation demonstrates that ScoreFormer not only achieves state-of-the-art accuracy in hit identification and docking score prediction but also offers a significant reduction in inference time compared to existing methods. Moreover, we also present L-ScoreFormer, a smaller version of the original model, focused on the general prediction of docking scores with improved efficiency. Both approaches make efficient and reliable tools for HTVS, capable of handling diverse datasets and conditions. The datasets, including scores, SMILES, and poses, used in our testing across multiple systems, with and without docking constraints, will be made available at Zenodo.

## 2. Related work

HTVS applied to relatively large libraries has been a key practice in drug discovery for some time. Within HTVS, tra-

[1]Department of Artificial Intelligence, Nostrum Biodiscovery S.L., Barcelona, Spain [2]Department of Drug Discovery, Nostrum Biodiscovery S.L., Barcelona, Spain [3]Electronic and Atomic Protein Modelling Group, Barcelona Supercomputer Center, Barcelona, Spain [4]Institució Catalana de Recerca i Estudis Avançats (ICREA), Barcelona, Spain. Correspondence to: Álvaro Ciudad <alvaro.ciudad@nostrumbiodiscovery.com>, Alexis Molina <alexis.molina@nostrumbiodiscovery.com>.

*Accepted at the 1st Machine Learning for Life and Material Sciences Workshop at ICML 2024.* Copyright 2024 by the author(s).

ditional methodologies, such as Glide (Friesner et al., 2004) and rDock (Ruiz-Carmona et al., 2014), employ specialized scoring functions and search-based protocols in their algorithms. These are designed to prioritize speed, enabling the assessment of larger compound collections than typically seen in standard docking campaigns. However, with the ever-increasing size of modern compound libraries, these traditional docking approaches are becoming less suitable for the task at hand.

Advancements in HTVS have been greatly influenced by the introduction of surrogate models. In a study conducted by Agrawal et al. (2019), a progressive docking approach incorporating molecular features and Morgan fingerprints achieved a 50-fold speed increase and a 90-fold enrichment compared to whole library docking and random sampling, respectively. Graff et al. (2021) and Graff et al. (2022) utilized Bayesian optimization and a design space pruning framework on the MolPAL platform, attaining a notable top-10k recovery rate of 67.7% while significantly reducing computational demands. Moreover, Yang et al. (2021)'s active learning framework, which integrates a Graph Convolutional Network(GCN)-based surrogate model, successfully identified 80% of experimentally confirmed hits, reducing computational costs by a factor of 14. These metrics emphasize the substantial enhancements in efficiency and accuracy these methods bring to HTVS.

GNNs have become increasingly effective in molecular learning due to their ability to model the complex, non-Euclidean structure of molecular data. Molecules, represented as graphs with atoms as nodes and bonds as edges, align well with GNN architectures, which effectively capture node interactions. The work by Hosseini et al. (2022) showcases the use of GNNs in enhancing docking score prediction, integrating machine learning-based surrogate docking with traditional methods to improve screening efficiency. Their introduction of FiLMv2, a novel GNN architecture, has shown remarkable performance improvements, including a 9.496-fold increase in molecule screening speed with respect to DOCK (Ewing et al., 2001) and a recall error rate below 3%, achieving significant speed increases and low recall error rates in molecule screening, thereby advancing chemical docking tasks.

However, the use of virtual nodes in GNNs, including FiLMv2, to capture global information can lead to increased computational complexity and prolonged training and inference times, presenting a challenge in HTVS. Additionally, virtual nodes risk overshadowing important local interactions and molecular topologies crucial for precise molecular property predictions due to the absence of learnable weights. To mitigate these issues, we propose the adoption of an attention mechanism, as an alternative to virtual nodes. This approach efficiently processes global information and re-

duces computational demands, while preserving the ability to capture essential local variations in molecular structures, as shown in the next section.

## 3. Methods

### 3.1. ScoreFormer Architecture

The ScoreFormer architecture is grounded in the graph transformer framework as introduced in Rampášek et al. (2022). This choice is motivated by the significance of long-range interactions in molecular regression tasks, a finding supported by ablation studies in Hosseini et al. (2022). Traditionally, a virtual node strategy from Gilmer et al. (2017) facilitates long-range interaction by introducing an extra node connected to every other node. However, our analysis (see Section 4.3) indicates that this approach substantially increases inference time. To mitigate this, we integrate an attention mechanism capable of modeling similar long-range interactions but with learnable parameters, resulting in enhanced performance.

To enhance the aggregation process from a node's neighborhood, in the message passing component of the graph transformer layers, we employ Principal Neighborhood Aggregation (PNA) as proposed by Corso et al. (2020). The PNA mechanism is defined as:

$$\mathbf{X}_i^{(t+1)} = \mathbf{U}^{(t)}\left(\mathbf{X}_i^{(t)}, \bigoplus_{(j,i)\in E}\left(\mathbf{M}^{(t)}\left(\mathbf{X}_i^{(t)}, E_{j\to i}, \mathbf{X}_j^{(t)}\right)\right)\right) \quad (1)$$

where $\mathbf{X}_i^{(t)}$ is the node feature matrix at layer $t$, $E_{j\to i}$ the edge feature of (j,i) if present, $\bigoplus$ are the set of aggregator and scalers defined in the original paper and $M^t$ and $U^t$ are neural networks at layer $t$.

Furthermore, to better capture positional information within the graph structure, we incorporate Learnable Random Walk Positional Encodings (LRWPE) from Dwivedi et al. (2022). These encodings are integrated at each layer of the PNA as follows:

$$\mathbf{x}_i^{(t+1)} = \text{PNA}\left(\mathbf{x}_i^{(t)} \oplus \mathbf{p}_i^{(t)}, E_{\mathcal{N}(i)\to i}, \mathcal{N}(i)\right) \quad (2)$$

$$\mathbf{p}_i^{(t+1)} = \text{PNA}\left(\mathbf{p}_i^{(t)}, \mathcal{N}(i)\right) \quad (3)$$

where $\mathbf{p}_i^{(t)}$ is the learnable positional encoding for node $v$ at layer $t$, $\oplus$ denotes concatenation as the operation for combining node features and positional encodings and $\mathcal{N}(i)$ is defined as the neighbouring nodes of $i$. The positional encodings are learned layer-wise with another PNA component.

The LRWPE are optimized simultaneously with the GNN parameters through a task-specific loss function, ensuring that both the feature and positional information are appropriately leveraged to improve the model's performance on molecular regression tasks. This dual optimization is a key feature of our architecture, distinguishing ScoreFormer from traditional graph transformer models by providing a more nuanced and positionally-aware representation of graph-structured data. A schematic representation of the ScoreFormer architecture is shown in Figure 1.

For improved efficiency, we also developed L-ScoreFormer, a streamlined variant of the ScoreFormer architecture, designed for efficiency and reduced overfitting risk. Developed with an emphasis on minimizing computational costs while maintaining performance, this model features a parameter-reduced structure. Its optimization, achieved through multiple objective trials using Optuna (Akiba et al., 2019), focuses on balancing parameter reduction with model efficacy. This approach ensures that L-ScoreFormer retains the core capabilities of the original ScoreFormer, yet operates more efficiently and with a lower risk of overfitting to specific data distributions.

### 3.2. Datasets

Our evaluation of ScoreFormer and FiLMv2 involved seven datasets, including one from the ZINC database with 128 million molecules docked to the dopamine D4 receptor (Lyu et al., 2019). We further utilized six datasets from the ZINC 20 database (Irwin et al., 2020), choosing 500k molecules with molecular weights between 200 and 500 Dalton. These molecules were docked against three protein systems, Dopamine D4 receptor, HIV-1 protease, and Cyclin-dependent kinase 2 (CDK2), using the Glide software. To assess model performance in different scenarios, we applied both constrained and unconstrained docking protocols, offering a comparative analysis of more realistic and idealized docking conditions. Further details are provided in Appendix A.

### 3.3. Model evaluation

In the following experiments, model performance was assessed by training the model using an 80-10-10 data split protocol. Inference epochs were selected based on F1 performance in a validation set composed of 10% of the dataset. Final performance metrics are obtained on a holdout test set using the remaining 10% of the dataset. This procedure was repeated for each of the protein systems and each docking protocol.

Performance was measured using the metrics employed in Hosseini et al. (2022), along with additional metrics for a more comprehensive comparison. Further details are in Appendix C.

## 4. Experiments

In evaluating both ScoreFormer architectures, our approach was three-folded, focusing on its generalization capabilities for a range of docking scores, efficacy in identifying promising drug candidates, and computational speed relative to leading models in the field. We utilized standard regression metrics to assess its predictive accuracy and specialized metrics for its precision in identifying potential high-affinity compounds. Additionally, we compared its processing speed against the currently available top-performing architecture. We also evaluated the generalization capabilities of ScoreFormer by performing inference on out-of-distribution molecules. Finally, we further conducted ablation studies to assess the contributions of LRWPE and PNA convolution layers to the performance of ScoreFormer, see Appendix G.

### 4.1. Prediction of docking scores

In implementing a graph transformer model like Score-Former, one of our primary objectives was to achieve superior generalization in predicting docking scores. This capability can be relevant in drug discovery campaigns, where there is a significant interest in compounds across different ranges of docking scores. Such comprehensive predictive ability is beneficial not only for HTVS tasks but also for parallel applications like Quantitative Structure-Activity Relationships (QSAR) (Hansch & Fujita, 1964).

To this end, we evaluated ScoreFormer's performance using standard regression metrics. According to the results in Appendix D, L-ScoreFormer demonstrates enhanced predictive power across a broad spectrum of docking scores, surpassing FiLMv2 implementation in both $R^2$ and Pearson. We hypothesize that this improvement is attributable to a reduction in overfitting, a common challenge in models trained with a focus on specific topologies, particularly under the WMSE loss used during training. This diminished tendency for overfitting in L-ScoreFormer suggests that a leaner model structure can be more effective in generalizing across different molecular structures and docking scenarios.

### 4.2. Molecular hit recovery

ScoreFormer's ability to recover molecular hits, crucial for its application in prospective HTVS campaigns, was thoroughly evaluated using the metrics stated in Appendix C. The analysis, as illustrated in Table 1, demonstrated Score-Former's superior performance in accurately identifying and prioritizing compounds with high binding affinity across various datasets and conditions. This proficiency is especially evident when comparing its performance to FiLMv2, a benchmark model in the field. ScoreFormer consistently outperformed FiLMv2 across multiple recovery metrics. Due

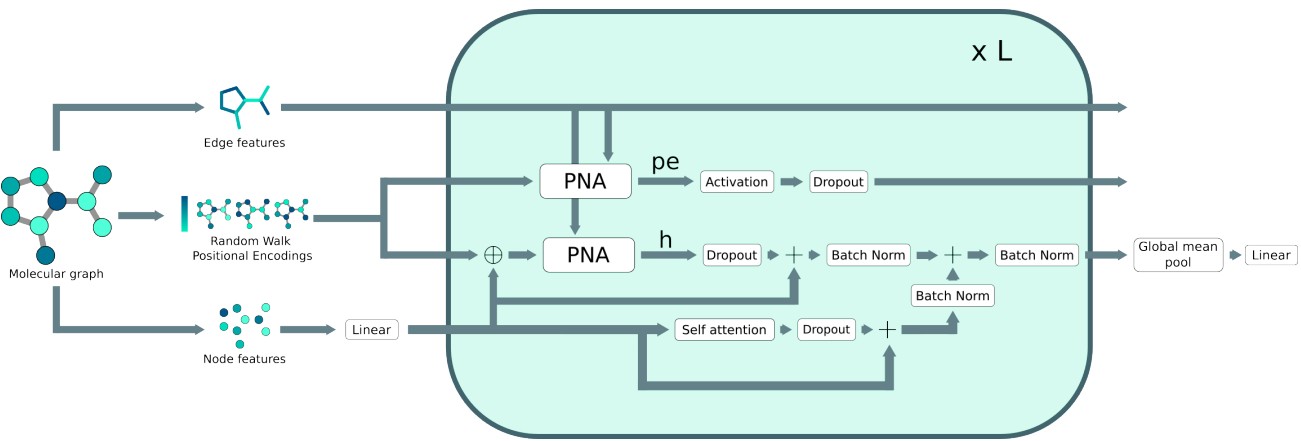

*Figure 1.* Schematic representation of the architecture used in the ScoreFormer model.

to its increase in the number of parameters, the model is able to capture the topological features responsible for good docking scores along the different systems and constrain set. These results highlight ScoreFormer's robustness and reliability in hit recovery. Notably, L-ScoreFormer, also displayed commendable performance, often closely following the FiLMv2 architecture. This observation underscores the effectiveness of both of our architectures in molecular hit recovery tasks. The consistent results of both ScoreFormer and L-ScoreFormer across diverse screening scenarios reinforce the versatility of our approach. See further results at Appendix E.

### 4.3. Inference speed

The inference speed of ScoreFormer, a critical factor in HTVS, was evaluated using a dataset of 500,000 molecules from the DOCK database. Conducted on a single A30 GPU, this assessment was tailored to reflect the real-world demands of computational drug discovery. To focus purely on the model's core computational capabilities, we excluded pre-processing steps from the evaluation, as these are commonly cached in practical applications, for both Score-Former and the FiLMv2 model.

The results, as shown in Table 2, highlight a significant enhancement in L-ScoreFormer's processing efficiency compared to FiLMv2. Achieving a 1.86-fold increase in speed, L-ScoreFormer processed samples at a rate of 2468.4 per second, effectively halving the estimated processing time for 128 million samples (DOCK dataset) to 14.40 hours from FiLMv2's 26.85 hours. ScoreFormer, also outperformed previous benchmarks by a factor of 1.652. This marked speed improvement is not achieved through a reduction of trainable parameters but by the replacement of the virtual node in FiLMv2 by an attention mechanism. This highlights

ScoreFormer's architectural efficiency while demonstrating the model's capacity to handle large-scale datasets rapidly.

### 4.4. Generalization across the chemical space

To evaluate the generalization capabilities of ScoreFormer, we conducted two benchmarks with molecules outside the training distribution. These benchmarks are necessary due to the combinatorial nature of the training databases, which may lead to an unfair assessment of the generalization capabilities of this kind of methods.

The first benchmark uses molecules generated by generative models for the evaluation set. As shown in Filella-Merce et al. (2023), generative models can escape densely populated patent spaces, such as the one of CDK2. This results in a set of molecules with novel chemistry, which can be used to evaluate chemical motif generalization. The evaluation set included molecules with a maximum tanimoto similarity of 0.3 to the training dataset of the generative model. Docking scores were calculated using the same constrained protocol reported in this study and we evaluated all three previous models, comparing them with metrics obtained by ScoreFormer on non-generated molecules. All metrics showed a decline, but ScoreFormer and L-ScoreFormer performed better or equal to FiLMv2, demonstrating better generalization capabilities for out-of-distribution molecules. [1] Results are summarized in Table 3 and full results can be found in Appendix H.

The second benchmark focuses on creating custom training and testing splits based upon molecular weight, to assess

---

[1] Due to low overlap of the new docking scores with the docking scores of the original training distribution, we do not report the regression metrics values, as all the models under-perform in this task. However, we analyzed molecular hit recovery capability, as this can still be inferred from available results.

*Table 1.* Metrics for FiLMv2, ScoreFormer, and L-ScoreFormer for the seven datasets chosen. Best performance metrics are highlighted in **bold**.

| TARGET | ENGINE | SETTINGS | MODEL | WMSE | RES | $AURTC_{0.01}$ | $AURTC_{0.001}$ | $R_{0.1,0.01}$ | $R_{0.1,0.001}$ |
|---|---|---|---|---|---|---|---|---|---|
| D4 | DOCK | UNCONSTRAINED | FiLMv2 | 0.399 | 0.754 | 0.644 | **0.721** | **0.886** | **1.000** |
| | | | ScoreFormer | **0.388** | 0.755 | 0.646 | 0.715 | **0.886** | 0.976 |
| | | | L-ScoreFormer | 0.398 | **0.763** | **0.647** | 0.691 | 0.874 | 0.952 |
| | GLIDE | UNCONSTRAINED | FiLMv2 | 0.383 | 0.813 | 0.706 | **0.802** | 0.901 | **1.000** |
| | | | ScoreFormer | **0.358** | **0.820** | **0.721** | 0.789 | **0.909** | **1.000** |
| | | | L-ScoreFormer | 0.389 | 0.817 | 0.711 | 0.764 | 0.868 | **1.000** |
| | | CONSTRAINED | FiLMv2 | 0.349 | 0.817 | 0.760 | 0.749 | 0.912 | **0.951** |
| | | | ScoreFormer | **0.321** | **0.845** | **0.795** | **0.803** | **0.924** | **0.951** |
| | | | L-ScoreFormer | 0.344 | 0.826 | 0.763 | 0.750 | 0.905 | **0.951** |
| CDK2 | GLIDE | UNCONSTRAINED | FiLMv2 | **0.602** | **0.707** | **0.613** | **0.655** | 0.760 | 0.875 |
| | | | ScoreFormer | 0.603 | 0.698 | 0.603 | 0.624 | **0.762** | **0.917** |
| | | | L-ScoreFormer | 0.629 | **0.707** | 0.599 | 0.622 | 0.738 | 0.875 |
| | | CONSTRAINED | FiLMv2 | 0.510 | 0.737 | 0.632 | 0.702 | 0.801 | 0.935 |
| | | | ScoreFormer | 0.519 | 0.746 | 0.653 | 0.735 | **0.826** | 0.968 |
| | | | L-ScoreFormer | **0.508** | **0.758** | **0.675** | **0.744** | 0.804 | **1.000** |
| HIV | GLIDE | UNCONSTRAINED | FiLMv2 | 0.629 | 0.674 | 0.537 | 0.643 | 0.690 | 0.918 |
| | | | ScoreFormer | 0.640 | 0.691 | **0.562** | **0.695** | **0.712** | **0.980** |
| | | | L-ScoreFormer | **0.614** | **0.696** | 0.559 | 0.682 | 0.675 | 0.939 |
| | | CONSTRAINED | FiLMv2 | 0.450 | 0.711 | 0.588 | 0.657 | 0.749 | 0.875 |
| | | | ScoreFormer | **0.439** | **0.725** | **0.599** | **0.678** | **0.771** | 0.875 |
| | | | L-ScoreFormer | 0.455 | 0.711 | 0.580 | 0.644 | 0.719 | 0.800 |
| ALL | ALL | ALL | FiLMv2 | 0.475 | 0.745 | 0.640 | 0.704 | 0.814 | 0.936 |
| | | | ScoreFormer | **0.467** | **0.754** | **0.654** | **0.720** | **0.827** | **0.952** |
| | | | L-ScoreFormer | 0.477 | **0.754** | 0.648 | 0.700 | 0.798 | 0.931 |

*Table 2.* Speed performance of the three models tested in this study. Speed is indicated as samples predicted per second and time needed for predicting 128 million docked compounds. The number of learnable parameters is also added for comparison.

| MODEL | SAMPLES/S | 128M TIME (H) |
|---|---|---|
| FiLMv2 | 1323.942 | 26.850 |
| ScoreFormer | 2186.828 | 16.259 |
| L-ScoreFormer | 2468.404 | 14.404 |

| MODEL | SPEEDUP | # PARAMETERS |
|---|---|---|
| FiLMv2 | 1.000 | 102977 |
| ScoreFormer | 1.652 | 5398273 |
| L-ScoreFormer | 1.864 | 147457 |

molecular size generalization, and its results can also be found at Appendix H.

## 5. Conclusions

In this study, we introduce ScoreFormer and its variant L-ScoreFormer, graph transformer models designed to overcome limitations in current approaches for surrogate models in HTVS. ScoreFormer's architecture, incorporating an attention mechanism, effectively manages global and local molecular information, leading to superior performance in identifying high-affinity compounds. L-ScoreFormer, with fewer parameters, shows strong performance in general docking score prediction, reducing overfitting risks. A key

*Table 3.* Metrics obtained on generated molecules. Reference configuration corresponds to constrained ScoreFormer results on CDK2 on Table 1. Best values, excluding the reference, are highlighted in **bold**.

| CONFIGURATION | RES | $AURTC_{0.01}$ | $AURTC_{0.001}$ |
|---|---|---|---|
| ScoreFormer | **0.458** | 0.344 | **0.359** |
| L-ScoreFormer | 0.449 | **0.358** | 0.336 |
| FiLMv2 | 0.431 | 0.333 | 0.314 |
| Reference | 0.746 | 0.653 | 0.735 |

feature is the improved inference speed, with a 1.86-fold increase, vital for high-throughput virtual screening. Future work will focus on applying ScoreFormer in active learning contexts, integrating explicability modules, and adopting uncertainty estimation methods, aiming to further enhance its utility in drug discovery and contribute to the advancement of computational chemistry with more reliable, efficient, and interpretable methods.

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

## A. Molecular docking in CDK2, D4 and HIV-1.

### A.1. System preparation

All systems were prepared with the same protocol and the structures were the following: CDK2 with PDB id 3BHV (Bettayeb et al., 2007), D4 with PDB id 5WIU (Wang et al., 2017) and HIV-1 with PDB id 3AID (Rutenber et al., 1996).

Protonation states were assigned at pH 7.4 using the Protein Preparation Wizard (Madhavi Sastry et al., 2013). Hydrogen bonds were optimized with PROPKA (Søndergaard et al., 2011) and a restrained minimization using the atom force field OPLS4 (Lu et al., 2021) was applied to each system.

### A.2. Glide docking

Ligand dockings were performed using Schrödinger's Glide software, we used the Standard Precision (SP) protocol with a fast sampling of 500 poses kept per ligand for the initial phase of docking, 40 best poses kept per ligand for energy minimization and only one pose selected for each compound after minimization. Docking grids were centered into the centroid of the corresponding native ligand with an inner box of 15 Å for D4 and HIV-1 and 10Å for CDK2. Hydrogen-bond constraints were imposed to improve the applicability of the poses generated. Constraints selected per system:

1. CDK2: hydrogen bond constraint to the central donor of the CDK2 hinge region (LEU83) (Li et al., 2015).

2. D4: hydrogen bond constraint to bias the selection of the compounds to selective agonists (ASP115) (Wang et al., 2017).

3. HIV-1: hydrogen bond constraint to the highly conserved aspartate in an aspartic protease (ASP25) active site to mimic ligand interactions of known inhibitors (Rutenber et al., 1996).

Docking was also performed on the unconstrained systems.

## B. Hyperparameters

Table 4. Hyperparameters for Scoreformer model.

| GROUP | PARAMETER | VALUE |
|---|---|---|
| TRAINING PARAMETERS | BATCH SIZE | 512 |
| | LEARNING RATE | 0.001 |
| | OPTIMIZER | ADAM |
| GPS PARAMETERS | ACTIVATION | RELU |
| | ACTIVATION PE | TANH |
| | DROPOUT | 0.100 |
| | DROPOUT ATTENTION | 0.500 |
| | DROPOUT PE | 0.100 |
| | NUM CONV LAYERS | 4 |
| | RESIDUAL WEIGHT | 1 |
| LPSE PNA PARAMETERS | AGGREGATORS | MEAN, MIN, MAX, STD |
| | HIDDEN DIM | 128 |
| | MLP ACTIVATION | NA |
| | MLP ACTIVATION PE | NA |
| | PRE AGGREGATION FC LAYERS | 1 |
| | POST AGGREGATION FC LAYERS | 1 |
| | SCALERS | IDENTITY, AMPLIFICATION, ATTENUATION |
| | TOWERS | 4 |
| | RW LENGTH | 9 |

*Table 5.* Hyperparameters for L-Scoreformer model.

| GROUP | PARAMETER | VALUE |
|---|---|---|
| | BATCH SIZE | 64 |
| | LEARNING RATE | 1.4126E-05 |
| TRAINING PARAMETERS | OPTIMIZER | ADAM |
| | ACTIVATION | ELU |
| | ACTIVATION PE | RELU |
| | DROPOUT | 0.245 |
| GPS PARAMETERS | DROPOUT ATTENTION | 0.432 |
| | DROPOUT PE | 0.188 |
| | NUM CONV LAYERS | 4 |
| | RESIDUAL WEIGHT | 0.035 |
| | AGGREGATORS | MUL, ADD, SUM, MEAN, MOMENT4, MOMENT5 |
| | HIDDEN DIM | 24 |
| | MLP ACTIVATION | ELU |
| | MLP ACTIVATION PE | RELU |
| | PRE AGGREGATION FC LAYERS | 3 |
| LPSE PNA PARAMETERS | POST AGGREGATION FC LAYERS | 2 |
| | SCALERS | LINEAR, INVERSE_LINEAR |
| | TOWERS | 1 |
| | RW LENGTH | 2 |

## C. Evaluation metrics

### C.1. Exponentially weighted mean squared error (W-MSE)

This metric was used as the loss function during training. It modifies a preexisting loss function, in this case, the mean squared error in Equation 5, to increase the weight of samples with low target values in Equation 4. In the context of molecular docking, we are specifically interested in accurately predicting the docking score of good binders, which are given lower values. This metric allows the training process to pay less attention to poor binders and hence becomes a more valuable tool for identifying good binders

$$\sum_{i=0}^{N} e^{-\alpha y_i} \cdot l(z_i, y_i) \tag{4}$$

$$l(z, y) = (z - y)^2 \tag{5}$$

### C.2. Regression enrichment surface score

This score corresponds to the volume under the surface defined as the recall of an estimator using two thresholds ($\zeta$, $\sigma$) for binarizing the target variable and the model outputs. The thresholds are expressed as the fraction of the samples to be considered as positive instances and hence the surface is bounded from 0 to 1 in both axes. The volume is computed with logarithmically scaled axes which results in the higher impact of samples with low target and predicted values in the overall metric.

### C.3. Area under recall threshold curve ($AURTC_\zeta$)

The recall threshold curve is defined as a slice of the RES by fixing the threshold $\zeta$ (the ratio of the model predictions to be considered positive). As in Hosseini et al. (2022), we report AURTC using two $\zeta$ values for all experiments: 0.01 and 0.001.

### C.4. Parametrized recall ($R_{\zeta,\sigma}$)

The parametrized recall is the fraction of retrieved positive examples by the model when using two thresholds i.e. $\zeta$ and $\sigma$, for dividing the true labels and the predicted labels into positive and negative examples. It corresponds to the value of the

RES at a given point. As in the original publication, we report R using two pairs of thresholds for all experiments: 0.1 and 0.01; 0.1 and 0.001.

## C.5. Other metrics

Beyond the metrics listed, model performance was also evaluated using the $R^2$ and Pearson's correlation coefficient.

## D. Regression results

*Table 6.* Correlation metrics for the different models and targets. Best performing models are highlighted in **bold**.

| TARGET | ENGINE | SETTINGS | MODEL | PEARSON | $R^2$ |
|---|---|---|---|---|---|
| D4 | DOCK | UNCONSTRAINED | FILMV2 | 0.725 | 0.466 |
| | | | SCOREFORMER | 0.736 | 0.472 |
| | | | L-SCOREFORMER | **0.750** | **0.522** |
| | GLIDE | UNCONSTRAINED | FILMV2 | 0.751 | 0.520 |
| | | | SCOREFORMER | **0.777** | 0.558 |
| | | | L-SCOREFORMER | 0.765 | **0.559** |
| | | CONSTRAINED | FILMV2 | 0.714 | 0.451 |
| | | | SCOREFORMER | **0.748** | **0.517** |
| | | | L-SCOREFORMER | 0.729 | 0.479 |
| CDK2 | GLIDE | UNCONSTRAINED | FILMV2 | 0.596 | 0.247 |
| | | | SCOREFORMER | 0.585 | 0.228 |
| | | | L-SCOREFORMER | **0.616** | **0.322** |
| | | CONSTRAINED | FILMV2 | 0.556 | 0.176 |
| | | | SCOREFORMER | 0.542 | 0.154 |
| | | | L-SCOREFORMER | **0.577** | **0.260** |
| HIV | GLIDE | UNCONSTRAINED | FILMV2 | **0.675** | 0.383 |
| | | | SCOREFORMER | 0.669 | **0.394** |
| | | | L-SCOREFORMER | 0.673 | 0.350 |
| | | CONSTRAINED | FILMV2 | 0.645 | 0.337 |
| | | | SCOREFORMER | **0.651** | **0.351** |
| | | | L-SCOREFORMER | 0.637 | 0.308 |
| ALL | ALL | ALL | FILMV2 | 0.666 | 0.368 |
| | | | SCOREFORMER | 0.673 | 0.382 |
| | | | L-SCOREFORMER | **0.678** | **0.400** |

# E. Performance plots

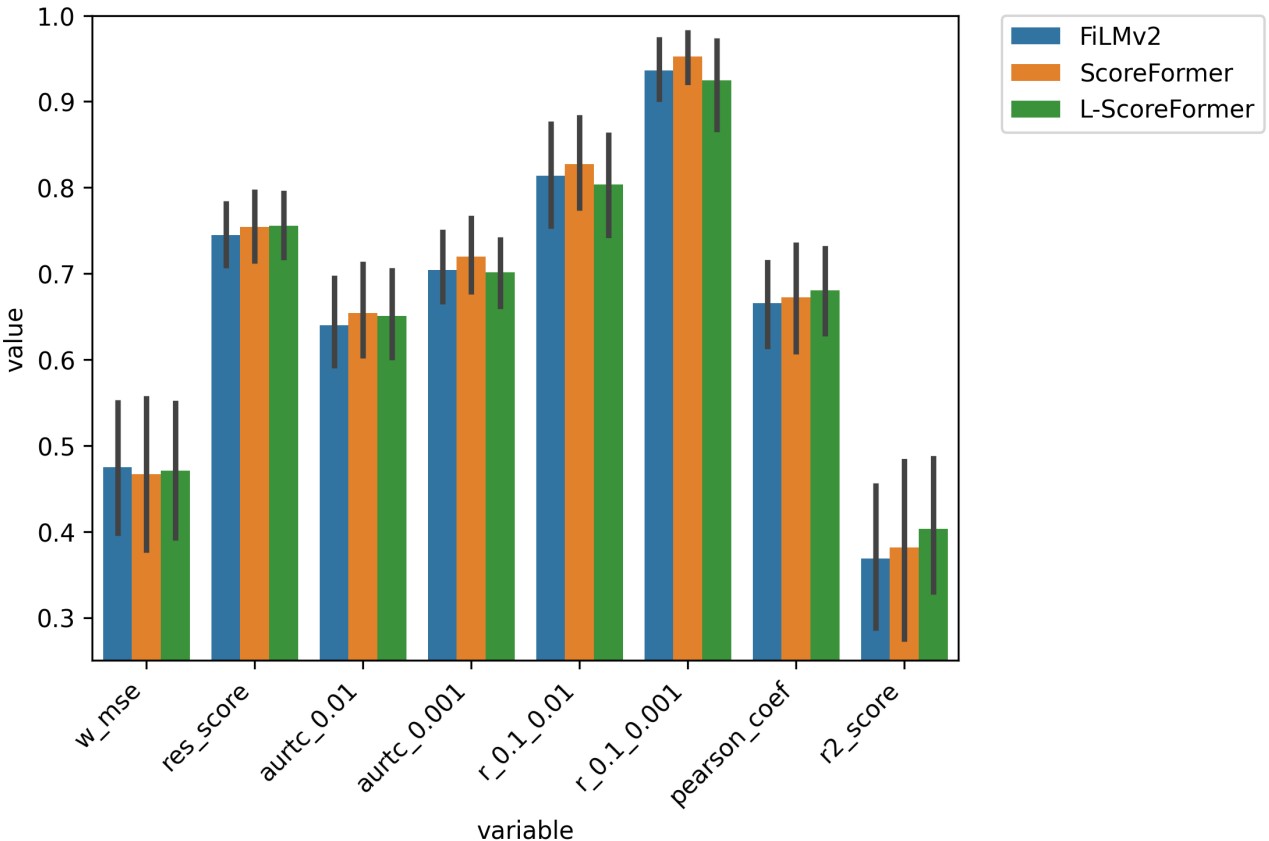

*Figure 2.* Performance metrics by model type. Bar height indicate the mean performance across different targets, docking engines and docking settings. The metrics reported are those used for the evaluation of docking score prediction and hit recovery.

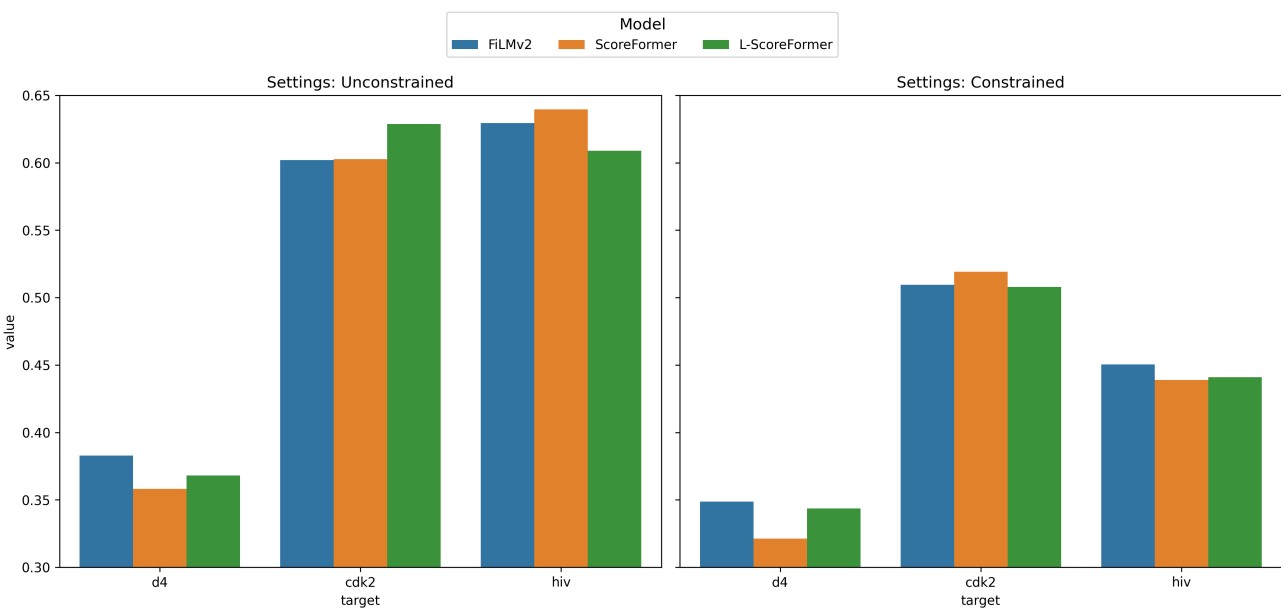

*Figure 3.* wMSE grouped by docking settings, target and model type. Data is only shown for the docking engine glide as is the only one presenting the two different types of docking settings.

# F. Pipeline

In this section, we describe the key features of the pipeline used to develop a functional surrogate model tailored to specific targets. An overview of the pipeline is shown in Figure 4. For each target of interest, a unique dataset is generated by calculating the docking scores for a set of molecules. These molecules are represented as molecular graphs, which encode atom types and their connections. Additionally, each atom's representation is augmented with the RWPE which stored for convenience. These graph representations, along with the docking scores, are used to train the model. Crucially, information about the target's structure, sequence, or the molecule's binding pose is not directly included as input. Instead, the model is designed to infer which molecular features are critical for binding affinity and how they influence this for a specific target. Employing a target-specific approach offers superior performance compared to target-agnostic models and, although it requires generating and training a new dataset and model for each target, it uses significantly fewer resources than conducting a full-scale HTVS. This makes target-specific pipelines an efficient balance between speed and precision.

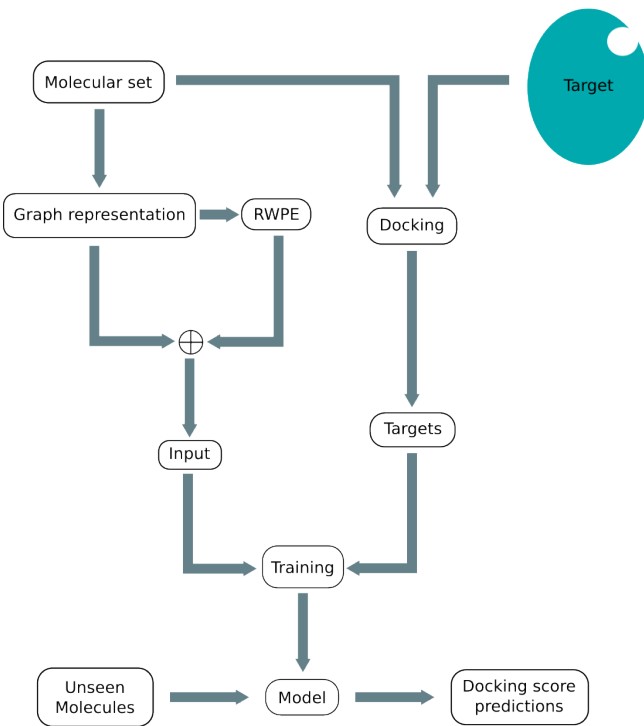

*Figure 4.* Schematic representation of the pipeline used to generate a functional model able to predict docking scores of new molecules.

# G. Ablation studies

To assess the impact of various components on model performance, we conducted three ablation studies. In the first study, we replaced the PNA network with two well-known GNNs, namely Graph Convolutional Network (GCN) and Graph Attention Network v2 (GATv2). In the second study, we removed the LRWPE from the ScoreFormer architecture. Lastly, we combined these two modifications, training a GPS graph transformer without LRWPE and using either GCN or GATv2 instead of the PNA network.

We evaluated these architectures using the D4 dataset, with docking scores calculated via Glide under the unconstrained protocol. The evaluation metrics included the average scores from the best 10 steps for each metric, as shown in Table 7, with the highest score for each metric highlighted in bold.

Our analysis focused on two metrics for score prediction — Pearson correlation and R² — and one metric for hit recovery, wMSE, evaluated on both the training and testing sets. Replacing the PNA layer with either GCN or GATv2 led to a decrease in performance across all analyzed metrics. Similarly, removing the LRWPE resulted in a comparable decline in all validation set metrics, though the wMSE for the training set remained unaffected. This indicates that LRWPE might have a regularization effect, helping to prevent overfitting and ensuring consistent performance across different datasets. Combining both ablations caused an even more significant performance drop across all metrics.

Collectively, these findings underscore the importance of the proposed components in enhancing the accuracy of docking score predictions.

*Table 7.* Metrics obtained by ScoreFormer and ablated architectures.

| MODEL | PEARSON | $R^2$ | TRAIN wMSE | TEST wMSE |
|---|---|---|---|---|
| SCOREFORMER | **0.779** | **0.564** | **0.269** | **0.341** |
| GCN | 0.762 | 0.531 | 0.309 | 0.364 |
| GATv2 | 0.761 | 0.526 | 0.297 | 0.364 |
| NO RWPE | 0.760 | 0.531 | **0.269** | 0.366 |
| NO RWPE GCN | 0.749 | 0.494 | 0.315 | 0.378 |
| NO RWPE GATv2 | 0.747 | 0.487 | 0.304 | 0.380 |

## H. Generalization studies

Here we include the complete table showing the results of the unseen chemistry benchmark using molecules obtained from a generative model.

*Table 8.* Extended metrics obtained on generated molecules. Reference configuration corresponds to constrained ScoreFormer results on Table 1. Best values, excluding the reference, are highlighted in **bold**.

| CONFIGURATION | RES | $AURTC_{0.01}$ | $AURTC_{0.001}$ | $R_{0.1, 0.01}$ | $R_{0.1, 0.001}$ |
|---|---|---|---|---|---|
| SCOREFORMER | **0.458** | 0.344 | **0.359** | 0.441 | 0.466 |
| L-SCOREFORMER | 0.449 | **0.358** | 0.336 | 0.441 | 0.466 |
| FILMV2 | 0.431 | 0.333 | 0.314 | 0.441 | 0.466 |
| REFERENCE | 0.746 | 0.653 | 0.735 | 0.826 | 0.968 |

Apart from the unseen chemistry benchmark, we conducted another which involved testing the generalization capabilities of our models across different molecular sizes. This test involved generating specific training and evaluation sets based on a molecular weight threshold and then inferring on the remaining testing molecules, with a higher molecular weight. Specifically, ScoreFormer was trained on the 90% of molecules with the lowest molecular weight and evaluated on the 10% of molecules with the highest molecular weight. To prevent any possible information leakage during evaluation, molecules in the validation set used for architecture optimization were excluded. The results, displayed in table 9 showed a slight degradation in performance compared to the model trained with a random selection of molecules. However, the model still captured feature patterns associated with binding affinities. The most significant degradation was observed in the weighted Mean Squared Error (wMSE) and the R-squared (R²) coefficient. Despite this, Pearson's correlation coefficient and metrics related to hit recovery were over 80% of those achieved by the random split approach, indicating that ScoreFormer can generalize to out-of-distribution molecules, at least, when there is a combinatorial relationship between the evaluation and training sets.

*Table 9.* Inference evaluation metrics on the molecular weight generalization benchmark.

| SPLIT | WMSE | RES | $AURTC_{0.01}$ | $AURTC_{0.001}$ | $R_{0.1, 0.01}$ | $R_{0.1, 0.001}$ | PEARSON | $R^2$ |
|---|---|---|---|---|---|---|---|---|
| WEIGHT | 0.697 | 0.719 | 0.610 | 0.656 | 0.753 | 0.917 | 0.627 | 0.33 |
| RANDOM | 0.383 | 0.813 | 0.706 | 0.802 | 0.901 | 1.000 | 0.751 | 0.52 |

Both of these benchmarks together allow us to evaluate the generalization capabilities of our models in various out-of-distribution situations and demonstrate significant robustness across them.