# OpenReview forum: "Scoreformer: A Surrogate Model For Large-Scale Prediction of Docking Scores"
_ICML.cc/2024/Workshop/ML4LMS — ML4LMS Poster_

### Official Review · Reviewer_gDpa · 2024-06-08

**Rating:** 5
**Confidence:** 3

**Review:**

This paper presented ScoreFormer, a GNN-based docking score prediction model for virtual screening purpose. ScoreFormer is composed of PNA as message passing scheme and incorporates random walk positional embedding for each node. The evaluation is based on the 7 datasets from ZINC database targeting different binding receptors.

Pros:

- mostly outperform previous baseline FILMv2 in the given benchmark tasks

Cons:

- Limited novelty/insights in aspect of model architecture and learning algorithm
- As in Tab2, the parameter size is far larger than FILMv2, yet in Tab2 the performance is not consistently better than baseline.

---

### Official Review · Reviewer_SoGD · 2024-06-12
**Scoreformer introduces a message passing GNN scheme for optimizing  high-throughput virtual screening scores**

**Rating:** 7
**Confidence:** 4

**Review:**

The authors want to capture long-range interactions in molecular regression tasks, for which they use a message passing graph neural network, with additional learnable positional encodings. They report the novelty as the simultaneous optimization of the GNN parameters and positional encoding parameters.

They evaluate the method with generalization capabilities for a range of docking scores, efficacy in identifying promising drug candidates, and computational speed.

The method itself is one which has components widely used used in deep learning architectures for biological data. While there may not be significant novelty there, it seems plausible that this a contribution to advancing the prediction and inference time for HTVS scores.